# Pathophysiological Concepts and Management of Pulmonary Manifestation of Pediatric Inflammatory Bowel Disease

**DOI:** 10.3390/ijms23137287

**Published:** 2022-06-30

**Authors:** Florian Schmid, Cho-Ming Chao, Jan Däbritz

**Affiliations:** 1Catholic Children’s Hospital Wilhelmstift, 22149 Hamburg, Germany; f.schmid@kkh-wilhelmstift.de; 2Department of Pediatrics, University Medical Center Rostock, 18057 Rostock, Germany; cho-ming.chao@med.uni-rostock.de; 3Cardio-Pulmonary Institute (CPI), University of Giessen and Marburg Lung Center (UGMLC), German Center of Lung Research (DZL), Justus-Liebig-University, 35398 Giessen, Germany; 4Department of Pediatrics, University Medical Center Greifswald, 17475 Greifswald, Germany

**Keywords:** children, molecular, inflammation, immunity, gut–lung axis, airways, pulmonary function tests

## Abstract

Pulmonary manifestation (PM) of inflammatory bowel disease (IBD) in children is a rare condition. The exact pathogenesis is still unclear, but several explanatory concepts were postulated and several case reports in children were published. We performed a systematic Medline search between April 1976 and April 2022. Different pathophysiological concepts were identified, including the shared embryological origin, “miss-homing” of intestinal based neutrophils and T lymphocytes, inflammatory triggering via certain molecules (tripeptide proline-glycine-proline, interleukin 25), genetic factors and alterations in the microbiome. Most pediatric IBD patients with PM are asymptomatic, but can show alterations in pulmonary function tests and breathing tests. In children, the pulmonary parenchyma is more affected than the airways, leading histologically mainly to organizing pneumonia. Medication-associated lung injury has to be considered in pulmonary symptomatic pediatric IBD patients treated with certain agents (i.e., mesalamine, sulfasalazine or infliximab). Furthermore, the risk of pulmonary embolism is generally increased in pediatric IBD patients. The initial treatment of PM is based on corticosteroids, either inhaled for the larger airways or systemic for smaller airways and parenchymal disease. In summary, this review article summarizes the current knowledge about PM in pediatric IBD patients, focusing on pathophysiological and clinical aspects.

## 1. Introduction

Inflammatory bowel disease (IBD) has shown an increasing prevalence in recent decades worldwide [1]. In 25% of patients, IBD manifestation occurs before the age of 18 [1]. Under the umbrella term of IBD, three different disease entities are subsumed: Crohn’s disease (CD), ulcerative colitis (UC) and inflammatory bowel disease—unspecified [2]. While CD potentially occurs in any part of the digestive tract from mouth to anus and shows an incontiguous, transmural inflammation, UC usually is an ascending, continuous inflammation beginning at the rectal mucosa. The currently accepted pathophysiological idea comprises a dysregulation of the enteric immune system in genetically predisposed individuals, which results in an inadequate response to commensal enteric bacteria [3]. However, IBD is not only understood as an intestinal, but also as a systemic disease, which can affect different organs, called extraintestinal manifestations (EIM) of IBD [4,5]. In the adult IBD population, approximately 30% develop EIM and a quarter of them even before the onset of IBD [6]. If one other organ beside the gut is affected, the likelihood of suffering from further EIM is significantly increased [7]. The prevalence of EIM in the pediatric population was reported to be from 17% to 28% after 15 years of diagnosis [8,9,10]. While 6% of pediatric patients develop EIM before the diagnosis of IBD, one fourth show EIM in the first 2 to 3 years after the diagnosis [10]. Almost ninety percent of EIM will occur in the first year of IBD course [9]. The most common EIM are arthritis, skin lesions (i.e., erythema nodosa and pyoderma gangrenosum), aphthous stomatitis, uveitis and hepatobiliary diseases (i.e., primary sclerosing cholangitis and hepatitis) [10,11,12]. The pulmonary system as a location of EIM is a rare condition (<1%), interstitial pneumonitis being seen more in UC patients and granulomatous disease being more associated with CD [13,14,15]. While in adults pulmonary manifestation (PM) is generally more detected in UC patients, most of the reported cases in children and adolescents are associated with CD [16]. Forty percent of adult IBD patients show respiratory alterations, which are frequently subclinical and might not be clinically relevant [12]. PM in pediatric IBD patients has rarely been described, mostly by case reports [17]. In a large cohort study of >1600 children with IBD, none showed PM [10]. However, it has to be noted that this number might be underrated, due to the fact that the majority of pediatric IBD patients with PM are asymptomatic. Furthermore, no regular screening for pulmonary disease has been established to date in the pediatric IBD population [18,19]. 

Two very informative review articles have already been published for either the clinical factors or pathophysiological concepts of PM in IBD [5,16]. Therefore, we aim to combine in our review article the most relevant aspects of clinical decision making in children with PM and the current knowledge about the immunological and molecular environment, which most likely leads to the development of PM.

## 2. Pathophysiological Concepts

Since the first report of a patient with IBD and pulmonary symptoms in 1976 by Kraft el al., several pathophysiological concepts for PM have been proposed [20].

### 2.1. Shared Embryological Origin

Most of the described links between pulmonary and intestinal tissue are based on the shared embryological origin in the foregut portion of the endoderm, leading to the possibility of a cross-mucosal interaction between the two organs that compose the human mucosal immune system [21,22]. Although much of this gut–lung interaction is not finally understood, both organs show a similar architecture, being composed of an extensive mucosal surface (with goblet cells and submucosal glands), containing lymphoid tissue (innate and adaptive immunity) and being covered by mucus, commensal bacteria and other antigens [16,23,24,25]. The gut and lung share the ability of a mucosal defense mechanism [17]. For instance, the dendritic cells of the lung are able to upregulate the expression of α4β7-integrine, which results in T lymphocyte (Tc) migration to the gut and therefore a direct communication pathway between the intestine and the lung [26]. Furthermore, due to the similar structure of these two mucosal organs, Papanikolaou et al. hypothesized that circulating autoantibodies, such as perinuclear anti-neutrophil cytoplasmic antibodies (pANCA) or anti-saccharomyces cerevisiae antibodies (ASCA), might lead to similar effects of inflammation in the lung as in the gut tissue [16].

### 2.2. Innate and Adaptive Immunity

In IBD, the homeostasis of the intestinal mucosal immune system is unbalanced, which results in a dysregulated response to luminal, commensal bacteria and other antigens [3]. The first step towards this dysregulation consists of a leaky intestinal mucus and interepithelial cell connection, which facilitates antigens to translocate through the intestinal epithelium [27,28]. In a second step, dendritic cells recognize these antigens as pathogens and induce a T-cell-mediated immune response [29]. Furthermore, activated macrophages induce the expression of leucocyte adhesion molecules at the endothelial blood vessels via the secretion of cytokines (interleukin 1 (IL-1) and tumor necrosis factor alpha (TNF-α)) [30]. This neutrophilic-mediated inflammation plays a key role in IBD development [5]. Neutrophilic cells (Nc) are generated from myeloid precursor cells in the bone marrow, induced by granulocyte colony stimulating factor and IL-17A and then released to the blood stream [31]. Passing by inflamed tissue, Nc become tethered via P-selectin glycoprotein ligand 1 to selectins (P-selectin and E-selectin) expressed by endothelium cells (EC) and consequently migrate into the inflamed tissue (termed margination and diapedesis) [32]. It was shown that the amount of margination and diapedesis of Nc in the pulmonary vasculature during systemic inflammation is augmented, which might be one reason for the trapping of neutrophils in the lung during active IBD [33]. Furthermore, in contrast to other organs, the neutrophilic emigration in the lung vasculature mainly takes place in smaller capillaries and not in post-capillary venules with a larger diameter. Therefore, the pooling time of Nc in the lung is prolonged [34,35]. Moreover, it was shown that pulmonary permeability is increased in patients with CD, facilitating the emigration of Nc [36]. Yipp et al. were able to demonstrate via confocal pulmonary intravital microscopy that, in bacteriemia, pulmonary Nc detect bacterial endotoxins (lipopolysaccharide) via Toll-like receptor 4 (TLR4) binding, resulting in the cluster of differentiation (CD)11b-dependent crawling of Nc. This finally leads to a quicker response to bacterial pathogens compared to the TLR4-myeloid-differentiation primary response gene 88 (Myd88) activation of macrophages. This activation hence triggers, via nuclear factor kappa B (NF-κB), the subsequent expression of TNF-α, chemokines and other proinflammatory molecules, apart from activation of EC to produce selectins for Nc migration [37]. In IBD patients, Nc become initially primed in the gut mucosa during episodes of disease flare. A subset of the these gut-primed Nc are able to relocate from the abluminal to the luminal membrane (reverse transendothelial migration (rTEM)) [38]. Furthermore, the inflamed gut tissue leads to an expression of IL-6, TNF-α, interferon-γ and vascular endothelial growth factor (VEGF) [39,40,41]. Elevated systemic levels of these factors result in an upregulation of ligands for extravasation of Nc and an increased vascular permeability in lung tissue [5,42]. Therefore, the primed (rTEM) Nc recirculate to the bloodstream and translocate into the respiratory endothelium [5]. In vivo studies reveale, that junctional adhesion molecule C is suppressed in inflamed, hypoxic tissue, leading to an increased rTEM of Nc. In addition, primed Nc change their shape and though become deformable, which results in a prolonged transit time through the pulmonary vasculature [38,43]. Interestingly, the healthy lung serves as an organ of de-priming Nc to stabilize the neutrophile homeostasis. In the injured lung, though, this function is compromised, leading to more primed Nc passing the pulmonary blood pool and therefore more Nc transmigrating via the arterial blood stream into the pulmonary mucosa. This finally leads to an exaggerated neutrophilic-mediated inflammation [44]. Aydin et al. found significantly increased VEGF and TNF-α concentration in lung tissue during experimental colitis in rats [45]. TNF-α not only leads to neutrophile accumulation and enhanced expression of adhesion molecules, but also to the formation of granuloma, resulting in the granulomatous manifestation of IBD in the lung [46]. Overall, it can be stated that the close relationship of the mucosal immune systems of the gut and lung via primed (rTEM) Nc might contribute to PM.

Tc represent a second cell type that exhibits a potential mechanism to promote pulmonary inflammation and subsequent tissue damage in the lung. In healthy organs, Tc tend to migrate to the same tissue, where they were first exposed to their specific antigen [47]. Unlike the specific attraction of gut memory Tc via the interaction of α4β7–integrin on Tc and mucosal addressin cell adhesion molecule 1 on intestinal EC, the migration of these Tc to the pulmonary system is based on an unspecific mechanism [48]. The non-tissue-specific binding of L-selectin/CD62L on Tc to peripheral lymph node addressin on EC and/or α4β1-integrin on Tc to vascular cell adhesion molecule 1 on EC results in the migration of lymphocytes to the bronchus-associated lymphatic tissue (BALT) [49]. It was shown that pulmonary Tc express further non-tissue-specific receptors termed C-C chemokine receptor 3 (CCR3) and C-X-C chemokine receptor 5 (CXCR5), which are both increasingly expressed in the Tc of inflamed intestinal tissue [50,51]. Therefore, memory Tc, which experienced their first contact to a specific antigen in the inflamed intestinal mucosa, carry a high number of CCR3 and CXCR5 and, as a result, can translocate to the BALT. This called “miss-homing” explains the presence of gut memory Tc in lung tissue, but it requires an initiation via a gut-specific stimulus to start the inflammatory process. The necessary antigens might derive from a breakdown of the intestinal mucosal barrier during IBD [52].

IL-25 is expressed by respiratory and intestinal epithelial cells and was shown to play an important role in the development of two inflammatory diseases—IBD and bronchial asthma [53,54,55,56]. Experiments in mice revealed that the neutralization of IL-25 results in a decreased inflammation of the colonic mucosa and diminished levels of pro-inflammatory cytokines, such as IL-1β, IL-6 and TNF-α [57]. However, in contrast, IL-25 levels are decreased in the colonic biopsies of CD patients and furthermore pro-inflammatory IL-13 is suppressed by the binding of IL-25 to a specific receptor at CD14^+^ monocyte-like cells [58]. Therefore, in IBD patients, IL-25 implies a chimeric pathogenic and protective function [59]. While in UC patients inflammation is trigged via T helper cells 2 (Th2), in CD, the expression of Th1-derived cytokines leads to intestinal inflammation [3]. IL-25 is mainly expressed by Th2 [60]. Therefore, it is not surprising that the pro-inflammatory effect of IL-25 is more evident in UC. IL-25 also plays a role in pulmonary inflammation upon allergen provocation [61]. To our knowledge, a specific link between the expression of IL-25 in PM of IBD was studied yet.

### 2.3. Further Molecular Concepts

The tripeptide proline-glycine-proline (PGP), formed from collagen by two enzymes (matrix metalloproteinase [MMP] and prolyl endopeptidase [PE]), has a chemotactic function for Nc in the intestine by binding to the chemokine receptor CXCR2 on neutrophils [62]. The expression of MMP and PE is elevated in intestinal inflammation. Therefore, more Nc are being attracted, which are able to express MMP and PE themselves [62]. This subsequently results in a neutrophil-mediated intestinal collagen proteolysis and therefore a vicious circle of neutrophilic inflammation [62]. This was shown earlier for inflammatory diseases of the lung, such as chronic obstructive pulmonary disease and asthma, and though it might be one cause of PM of IBD, to date no studies regarding the concentration of PGP in sputum of IBD patients have been published [63].

The already mentioned damage to the epithelial barrier seen in IBD leads to significant bacteremia [64]. This in turn results in neutrophil accumulation in the lung tissue, which is positively associated with levels of IL-1β and C-C-ligand 2 (CCL2), which play a role in pulmonary inflammatory diseases (i.e., bronchiectasis and chronic bronchitis) [65]. Liu et al. discovered, in murine models, that this neutrophil recruitment in the lung is triggered by the expression of platelet-activating factor receptor (PAFR), which then results in the overexpression of the inflammasome via nucleotide-binding oligomerization domain-containing protein (NOD)-like receptor family pyrin-domain-containing 3 (NLRP3) [66]. Therefore, not the actual bacterial colonization, but the activation of lung-hosted inflammasome might play a role in the development of PM [66]. Furthermore, the correlation between an enhanced expression of PAFR and diminished levels of L-selectin on Nc contributes to the trapping of Nc on the endothelial surface and hence the further activation of the inflammation processes [67].

Liu et al. were further able to demonstrate that inflammatory damage to the mucosa and microvascular endothelial changes, such as increased angiogenesis, adhesion molecule expression for lymphocytes (i.e., intercellular adhesion molecule 1), leukocyte extravasation and decreased endothelial barrier function, are similar in the intestinal and pulmonary mucosa during UC [68]. Thromboxane B2 is increasingly expressed in the lung and colonic tissue of murine UC models, which augments leukocyte recruitment and results in mucosal injury via the activation and aggregation of platelets, VEGF-A, which leads to angiogenesis, and finally inducible nitric oxide synthase (iNOS) enhancing leukocyte adhesion [68].

### 2.4. Microbiome

It seems inevitable that the intestinal microbiome plays an important role in the disease development of IBD [69]. Microbial exposure during early life not only decreases the probability of developing IBD later in childhood or adulthood, but also of other chronic inflammatory diseases occurring in pediatric patients, such as asthma [70,71]. Studying sensitization to food allergens, Stefka et al. discovered that fewer *Clostridia* spp. result in a dysregulated homeostasis of the host–commensal relationship via reduced IL-22 expression. This finally leads to the alteration of the epithelial junction integrity and the secretion of epithelial antimicrobial peptides [72]. Olszak et al. were able to demonstrate that invariant natural killer T cells accumulate in the lamina propria of the murine gut as well of the lung tissue after contact with commensal microbes early in life. This accumulation is associated with an elevated expression of chemokine ligand CXCL3, which then leads to more lymphatic migration and consequently to IBD or asthma [73].

### 2.5. Genetic Factors

The concept of IBD as an “organ barrier disease” might underpin a further connection between two organs with a large mucosal surface—the intestine and the lung. Around 40% of CD patients show disease-associated variants of the caspase recruitment domain-containing protein 15 (CARD15) gene, coding for the intracellular NOD2 receptor [74]. These mutations cannot be stated as alone standing factors for their contribution to IBD development, as the genetic susceptibility involvement is thought to be around 20% [75]. The NOD2 receptor, which is not only present in macrophages and dendritic cells, but also in intestinal epithelial cells, is part of a family of pathogen recognition receptors (PRR), recognizing muramyl dipeptide (MDP) as part of the bacterial cell wall [76,77]. The MDP-NOD2-mediated pathway leads via NF-κB activation and translocation to the nucleus to an enhanced expression of α-defensins. In 2004, Wehkamp et al. first discovered that the mutation of NOD2 in patients with CD results in a diminished expression of these α-defensins leading to a breakdown of the mucosal barrier [78]. Therefore, it can be hypothesized that in IBD patients with PM the NOD2 mutation might not only lead to altered inflammation in the gut, but also on the lung surface, especially because NOD2 mutations were also described in the pathogenesis of another inflammatory pulmonary disease—chronic obstructive pulmonary disease (COPD) [79]. Mutations in the autophagy-related 16-like 1 (ATG16L1) gene lead to a diminished function of Paneth cells, a defective antigen presentation and proinflammatory cytokine secretion in IBD patients [80]. ATG16L1 function in PM was not studied to our knowledge, so a concluding assessment, whether the mutation is another link between the intestine and the lung, cannot be made. While specific expressions of humane leucocyte antigen (HLA) are associated with the development of EIM, such as HLA-A2 and HLA-DR1 in CD and HLA-B27, HLA-B8/DR3 and HLA-B58 in UC, no linkage to HLA-types were found for PM [16,81,82].

Figure 1 summarizes the current knowledge about potential cellular and molecular mechanisms involved in the development of PM in patients with IBD.

## 3. Clinical Aspects

PM in pediatric IBD patients is, due to low incidence and low awareness in clinical settings, a rarely considered aspect. The most important clinical issues will be discussed here with a molecular emphasize.

### 3.1. Clinical and Radiological Findings

The severity of PM can vary from asymptomatic (detected only by pulmonary function tests (PFT)) to subclinical and severe [83]. Symptoms can be subdivided into those of the upper and lower respiratory tract, interstitial lung disease and pleural affections [84]. In the adult IBD population, the most affected compartment is the bronchial system leading to bronchiectasis or chronic bronchitis, even being diagnosed post-colectomy in patients with UC [25,85,86,87]. Colectomy might constitute a risk factor for the development of PM, explained by a molecular mimicry mechanism, where the pulmonary system is being attacked by earlier gut-sensitized lymphocytes, which lost their target after colectomy [86,88,89,90]. Adult patients present mainly with dyspnea, hoarseness, stridor or chronic coughing [85]. In several studies, the rates of respiratory symptoms in IBD patients were reported from 44 to 50% [91,92,93]. Over 40% revealed abnormal findings on thoracic high resolution computed tomography (HRCT), mainly centrilobular nodules and bronchial wall thickening [94]. Other frequently detected alterations in HRCT include bronchiectasis, reticulonodular opacities, emphysema and ground-glass alterations [12,95].

In the pediatric population, PM generally remains clinically silent and affects primarily the parenchyma (alveolar and interstitial), with organizing pneumonia being the most commonly described pathology [25,96]. On chest X-rays of patients with suspected interstitial involvement, focal or diffuse peripheral predominant opacities and air bronchogram can be detected [12]. HRCT, which is rarely used in children, shows patchy, asymmetric foci of consolidation in a peripheral or peribronchovascular distribution and ground-glass opacities [12]. Looking at radiographic diagnostics in children, Peradzynska et al. reported that HRCT identified only 1 out of 32 pulmonary asymptomatic patient with bilateral bronchiectasis [97]. In most of the published cases of children with IBD and pulmonary symptomatic PM, X-rays showed alterations (Table 1 and Table 2). If the airway system is affected in children with IBD, the inflammation concentrates on smaller airway compartments, leading to purulent bronchitis [18,86,98]. Adult IBD patients with small airway involvement tend to be younger and the proportion of PM, which occurs before intestinal symptoms, is higher than in those patients with large airway involvement [96]. The difference between pathological findings in HRCT and symptoms in children and adults emphasizes that pulmonary involvement in IBD shows a progressive nature, which hardly leads to clinical issues in childhood, but contributes to morbidity in adults.

Table 1 and Table 2 summarize all 34 case reports of PM in pediatric patients with CD (Table 1) and UC (Table 2) to date. While in the adult population PM is more frequently associated with UC, most of the reported pediatric PM cases were children with CD (74%) [87].

### 3.2. Pulmonary Function and Breathing Tests

PFT play an important role in monitoring children with chronic diseases of the respiratory system, although they are not yet established for screening for pulmonary involvement in children with IBD on a regular basis. Studies with adult patients reported a compromised lung function in up to 94% of patients with active IBD, particularly of the small airways [91,119,120,121,122,123,124]. While spirometry and plethysmography measurements in pediatric IBD patients showed no significant pathological alterations in a number of studies; the lung clearance index (LCI), as a possible marker of early damage and ventilation inhomogeneity, was significantly decreased [97,125,126,127]. This observation can lead to the hypothesis that early damage of the lung in pediatric IBD patients, which is not yet showing any clinical symptoms, might lead over time to more severe damage in adulthood [125].

A recent study showed alterations in certain parameters of PFT (forced expiratory volume in 1 s (FEV1), forced vital capacity and diffusion capacity of lung for carbon monoxide (DLCO)) between the day of IBD diagnosis and three years after [128]. Contradictory to other studies, Furlano et al. showed that pediatric CD patients have significantly decreased mid- and end-expiratory flow values, but the number of patients with flow values under the norm was generally low [98].

If the impaired gas transfer, measured via DLCO, might be a more reliable parameter to objectify the involvement of pulmonary interstitium, in pediatric IBD patients this is likewise not certain, as several studies revealed that pediatric IBD patients showed decreased DLCO levels, whereas other authors could not find any significant correlation [97,98,125,127,129,130].

NOS2 is induced by inflammatory stimuli, leading to a proinflammatory environment [131]. It was shown that the gene coding for NOS2 (NOS2A) is overexpressed in IBD, leading to increased nitric oxide levels in the colonic and rectal gas of UC patients by the factor 100 [132,133]. Therefore, it can be hypothesized that in IBD functional exhaled nitrite oxide (FeNO) levels might be elevated even earlier than PFT can detect airway inflammation. A study of Gut et al. showed that FeNO was elevated in children with IBD (CD and UC) compared to healthy individuals, independently from disease activity [134]. Three other studies with pediatric CD patients do not support this correlation [97,135,136]. In the adult population, the levels of FeNO seem to be associated with the disease activity of CD and UC [137,138].

Alternative methods of non-invasive monitoring of pulmonary inflammation can be obtained via exhaled breath condensate (EBC) samples. A decreased pH in EBC samples of pediatric CD patients, a sign of an inflammatory process, showed significantly lower levels than in healthy controls and in patients with asthma [136]. Furthermore, the correlation to disease activity converged significancy [136]. Other biomarkers, such as 8-isoprostane, a free-radical induced lipid peroxidation product of arachidonic acid, are elevated due to oxidate stress damage (such as in children with asthma) and were surprisingly lower than in the control group, which might be due to different oxidate stress pathways in CD patients [136,139]. The levels of trefoil factor 2 (TTF2) in EBC, a member of a group of growth-factor-like peptides, which are associated with mucosa barrier protection, were, unlike those in asthma patients, significantly decreased [136]. This is consistent with previous studies, which found lower levels of TTF2 in the intestinal mucosa biopsies of pediatric IBD patients than in non-inflamed mucosa samples [140]. Another study group looked at proinflammatory cytokines in EBC (IL-6, IL-8, IL-1β, and TNF-α) and found elevated levels for all parameters in pulmonary asymptomatic pediatric IBD patients with normal PFT [127]. This shows that monitoring of proinflammatory cytokines in EBC might be a further method to detect early inflammatory progress in lung tissue. 

There are two studies looking at bronchial hyperresponsiveness (BHR) in pediatric CD patients using methacholine challenge. In a small cohort, 71% of pulmonary asymptomatic pediatric CD patients showed BHR, with no correlation to disease activity, duration of disease course or treatment [141]. Livnat et al. reported 30% of pediatric CD patients with BHR [135]. Whether subclinical BHR is generally important for evaluation of pediatric IBD patients remains unclear.

### 3.3. Medication-Associated Lung Injury

Pulmonary symptoms in IBD should not be attributed to PM alone, as they can also be caused by medication [142]. Current medical treatment options for pediatric IBD include—beside exclusive enteral nutrition—5-ASA (mesalamine (MES), sulfasalazine (SUF)), immunosuppressives (i.e., prednisolone (PRD) and tacrolimus), immunomodulators (i.e., azathioprine (AZA) or methotrexate (MTX)) and therapies with biological agents (i.e., infliximab (IFX) and adalimumab (ADA); off-label-use in children: vedolizumab (VED), ustekinumab (UST), and tofacitinib) [143,144]. Medication-associated lung injury in pediatric IBD patients is scarce, but plays an important role in the management of pulmonary symptoms, as it is accountable for most non-infectious lung diseases in IBD patients [145]. While the medications used in IBD treatment usually affect more frequently the parenchyma, PM in adult patients are more likely to affect the airways. Therefore, differentiation between these two entities of lung disease might be more reliable than in childhood, where most PM likewise affect the lung parenchyma [142].

The treatment of IBD with MES and SUF are well recognized to cause lung injury with histopathological correlates, such as bilateral interstitial lymphocytic infiltrates, eosinophilic pneumonia, non-necrotizing pulmonary granulomatosis, bronchiolitis obliterans or fibrosing alveolitis [146,147,148]. MES seems to be associated with less pulmonary side effects than SUF [146].

While 25% of IBD patients receiving MTX showed pulmonary symptoms (i.e., cough, wheezing, and dyspnea), only 2–7% of patients will develop pneumonitis within several weeks of initiation [149,150,151].

Lee et al. studied over 1,000 adult IBD patients and reported six cases with anti-TNF-α-induced lung injury [152]. Before initiating therapy with TNF-α antagonists (i.e., IFX or ADA), it is obligatory to exclude active or latent infection with mycobacterium tuberculosis (TB) with Interferon-Gamma Release Assays and chest X-ray [153]. A cohort study of 873 adult IBD patients revealed 25 TB cases, of whom 76% developed TB despite an initially negative screening for latent TB [154]. Apart from TB, other infectious pulmonary side effects of IFX therapy were reported, almost invariably in adulthood (i.e., atypical mycobacterial infection, invasive fungal infections (such as histoplasmosis, aspergillosis or cryptococcosis), infection with pneumocystis jirovecii and nocardia) [155,156,157,158,159,160,161,162]. Review articles published by Tragiannidis et al. and Toussi et al. summarize the current knowledge regarding infections and particularly invasive fungal infections in children receiving TNF-α antagonists [163,164]. Non-infectious lung injury related to IFX include interstitial pneumonitis, alveolitis and diffuse alveolar hemorrhage, regularly resolving after drug discontinuation [165,166,167,168]. The pathogenesis is still unclear, but it was hypothesized that blocking proinflammatory TNF-α leads to a decreased apoptosis rate of inflammatory cells. These cells consequently remain in the pulmonary parenchyma, leading finally to tissue damage [169]. Recently, a case of a child experiencing VED-associated pulmonary disease was published, probably based on an integrin shift leading to proliferation of CD29^+^ lymphocytes (β1-integrin), which amplifies lymphocyte homing to the lung [170]. Mitchel et al. reported the case of a child with non-infectious bilateral pneumonia after the initiation of immunosuppressive therapy with UST [171]. To our knowledge, although reports of ADA-induced interstitial lung disease in adult IBD have been published, no pediatric cases have been mentioned to date [172]. A Cochrane analysis looked at fecal microbiota transplantation (FMT) in IBD patients and found some significance to achieve clinical remission in adult UC patients [173]. Two small randomized-control trials in CD patients showed a higher clinical remission rate versus placebo, with no reported adverse events [174]. However, pneumonia, as an adverse event using fecal microbiome transplantation, was reported in one UC patient [175].

### 3.4. Risk of Pulmonary Embolism

The incidence of venous thromboembolism (VTE) in children with IBD has been indicated to be between 0.6 and 1.9% [176,177]. The risk of pulmonary embolism (PE) is raised in children under the age of 16 in the first 5 years after the diagnosis of IBD [178]. A Danish registry study, looking at 5400 patients under 20 years of age, reported a PE incidence rate of 3.4 per 10,000 person-years in children with IBD compared to 0.6 per 10,000 person-years in the non-IBD cohort [179].

Nylund et al. found a relative risk (RR) of pulmonary embolism in CD of 1.8 in comparison to no-IBD children with hospitals stays, whereas UC patients did not show a significant difference [180]. The pathophysiology behind the raised risk of VTE in IBD is not well understood, but a number of alterations of the coagulation cascade and the fibrinolytic system were recognized [181]. Furthermore, persisting inflammation contributes to coagulation via cytokines, such as TNF-α, IL-6, IL-1β or C-reactive protein, which induce the coagulation cascade and inhibit anticoagulation pathways [182,183,184,185]. Recommendations for thromboprophylaxis in children with acute severe colitis and certain risk factors (i.e., smoking, oral contraceptives, complete immobilization and known thrombotic disorder) were implemented [186]. The current pediatric CD guideline of the European Crohn´s and Colitis Organisation (ECCO) and the European Society of Paediatric Gastroenterology, Hepatology and Nutrition (ESPGHAN) do not comment on thromboprophylaxis [143]. In a recently published article by the large European-based pediatric IBD registry named Paediatric Inflammatory Bowel Diseases network for Safety, Efficacy, Treatment and Quality improvement of care (PIBD-SETQuality), it was stated that venous thromboprophylaxis should be considered in all hospitalized children with moderate-to-severe CD with at least one additional VTE risk factor [187].

### 3.5. Treatment

Treatment options for PM in IBD depend on the localization and severity. While patients with upper airway inflammation, which can lead to severe airway obstruction (i.e., subglottic stenosis), should be treated with intravenous steroids, lower airway inflammation can be treated with inhaled glucocorticoids [87,90]. If upper airway inflammation is refractory to steroid treatment, interventional bronchoscopy can be performed (i.e., laser beam, balloon dilatation or stent placement) [16]. Furthermore, when PM is noticed, the escalation of IBD therapy is recommended [188]. In bronchiectasis, rare in children, antibiotic regimes and bronchial toilet should be applied [16]. The parenchymal involvement of PM, which is mostly seen in children, is best targeted with oral glucocorticoids [96,189]. When glucocorticoid therapy is not sufficient (which might be the case in one third of patients), IFX can be administered, if not yet part of the therapy management [190]. Krishan et al. reported three children with PM, who dramatically improved after the initiation of IFX treatment [105]. In cases of bronchiolitis, obliterans macrolides resulted in clinical improvement [16,191]. When medication-induced lung injury is considered, the offending drug has to be stopped and replaced [192].

## 4. Conclusions

The PM of IBD in children is an underestimated clinical problem. Most of the reported symptomatic pediatric cases covered CD patients. Current pathophysiological concepts of PM consist in the shared embryological origin of the gut and lung, a close connection between the pulmonary and intestinal immune system, the similar expression of distinctive proinflammatory molecules, genetic susceptibility and aspects of the microbiome. PM in children stays mostly silent, affecting more often the parenchyma than in the adult population. Due to contradictory study results in pediatric IBD patients, PFT are currently not playing a role yet in screening for PM, but measurement of LCI might show a promising way to detect early lung damage. Importantly, interstitial lung disease in children with IBD can also be caused by medications, such as mesalamine or biological therapies. PM can be generally treated with corticosteroids, either inhaled for the larger airways, or systemically for smaller airway involvement and parenchymal inflammation. When corticosteroids show no sufficient effect, IFX can be applied. Screening for PM in children with IBD should be considered a reasonable tool to prevent progressive lung damage in adulthood.

## 5. Methods

A systemic Medline search using the terms “lung”, “pulmonary”, “inflammatory bowel disease”, “Crohn’s disease”, “ulcerative colitis”, “IBD unclassified”, “pediatrics”, and “child” was performed. The databases searched included Cochrane, Embase and PubMed from April 1976 to April 2022. Articles not published in English language were excluded.

## Figures and Tables

**Figure 1 ijms-23-07287-f001:**
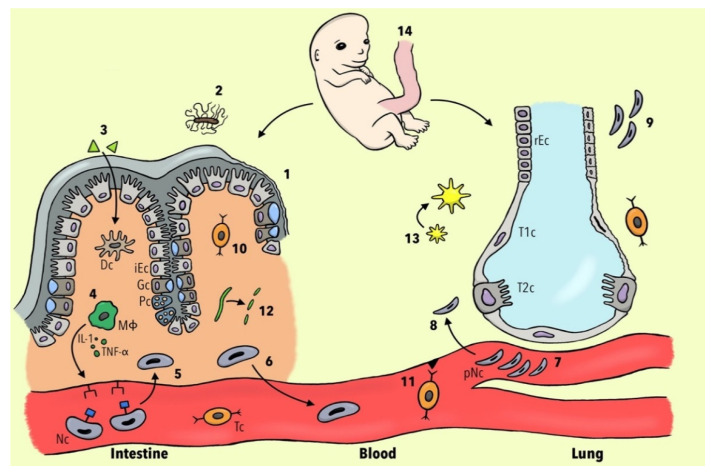
Different pathophysiological concepts of pulmonary manifestations in IBD patients. (1) Mutation of the CARD-15 gene leads to a diminished expression of α-defensins and consequently to a breakdown of the mucosal barrier. (2) Less *Clostridia* spp. result in dysregulation of host–commensal relationship and, via the reduction of IL-22, to the alteration of epithelial junction integrity. (3) Luminal antigens penetrate through the leaky intestinal mucosa and activate dendritic cells (DC). (4) Activated macrophages (ΜΦ) lead, via IL-1 and TNF-α, to the expression of neutrophils (Nc) adhesion molecules. (5) Nc undergo margination and diapedesis to the intestinal mucosa. (6) Nc become primed and a subset re-migrates to the bloodstream. (7) Nc change their shape and become deformable, resulting in a prolonged transit time in pulmonary capillaries. (8) JAM-C becomes downregulated and therefore the translocation of Nc to respiratory epithelium is facilitated. (9) The injured lung is not able to de-prime Nc leading to more neutrophilic inflammation. (10) Inflamed intestinal tissue leads to the upregulation of non-tissue-specific T-cell receptors CCR3 and CXCR5. (11) Diapedesis of gut-memory T cells (Tc)) via non-specific endothelial receptors to lung tissue is increased. (12) Collagen becomes degraded to tripeptide proline-glycine-proline, which plays a role in chemotaxis of Nc. (13) Increased bacterial burden leads to the expression of platelet-activating factor receptor and subsequently to the overexpression of the inflammasome. (14) Gut and lung tissues share the same embryological origin in the foregut portion of the endoderm. Abbreviations: iEC, intestinal epithelial cell; rEC, respiratory epithelial cell; Gc, goblet cell; Pc, Paneth cell; T1c, Type 1 pneumocytes; T2c, Type 2 pneumocytes; IL-1, interleukin 1; TNF-α, tumor necrosis factor alpha; pNC, primed neutrophils.

**Table 1 ijms-23-07287-t001:** Published cases of pulmonary manifestation in pediatric CD patients.

First Author	Publication Year	Age	Sex	δ	Pulmonary Symptoms	Radiographic Evaluation	PFT	Histopathology (Lung)	Therapy	Outcome
Ahmed [99]	2005	9y	F	Same time	Stridor, shortness of breath	XR: Normal BSC: Diffuse hemorrhagic tracheitis	Not performed	Tracheal biopsy: granulomatous inflammation	Not mentioned	Clinical improvement after 7mo of CD therapy
Akobeng [100]	1999	9y	M	4y (b)	Breathlessness, cough, wheeze	XR: Unilateral consolidation	Not performed	Not performed	Bronchodilators, antibiotics, physiotherapy	Little initial effect. No outcome mentioned after initiation of CD therapy
Al-Binali [11]	2003	11y	M	Same time	Exercise limitation, cough, shortness of breath	XR: Patchy airspace consolidation CT-T: Multiple peripheral nodules	Abnormal	Non-caseating epithelioid granulomatous inflammation	Not performed	Resolution of symptoms, radiological normalization and PFT improvement without therapy
Bentur [101]	2000	13y	F	10mo	Shortness of breath	XR: Normal CT-T: Patches of ground glass-appearing lung alternating with normal appearing areas of lung	Abnormal	Mononuclear inflammation with small non-caseating granulomas, bronchiolitis obliterans.	1. PRD, 6-MP 2. Hydroxy-chloroquine 3. After severe bone marrow depression switch to PRD + MES	Slow improvement of symptoms and PFT with combination therapy of PRD + MES
Calder [102]	1993	3y	M	Same time	None	XR: Unilateral densities	Not performed	Non-caseating epithelioid granulomatous inflammation	Not mentioned	Not mentioned
Chiaro [103]	2013	14y	F	4y	Non-productive cough, dyspnea, wheezing, chest pain	XR: Normal CT-T: Multiple pulmonary nodules, parenchymal infiltrates, diffuse bilateral interstitial inflammation	Not performed	Non-caseating granulomatous inflammation with multinucleated giant cells	Oral PRD	Clinical and radiological improvement after 2mo
Inoue [14]	2017	14y	M	14y	Shortness of breath, right pleuritic chest pain	XR + CT-T: Atelectasis, infiltration, pleural effusion	Not performed	Polypoid fibrosis, inflammatory cell infiltration	1. Intravenous Pulse-MP 2. Oral PRD	Complete resolution of symptoms soon after initiation of MP
Kayser [104]	1990	12y	M	3y	Tachypnea, dyspnea, increased mucus production	Not performed	Not performed	Granulomatous interstitial inflammation	1. Oral PRD 2. AZA	Improvement after initiation of AZA leading to improvement after 2y
Krishnan [105]	2006	13y	F	4y	Backpain, cough	XR: “Round pneumonia” CT-T: Multiple pleural and intraparenchymal infiltrates	Abnormal	Non-caseating epithelioid granulomatous inflammation with giant cells	1. Oral PRD 2. IFX	Immediate clinical and radiological response after initiation of IFX
Krishnan [105]	2006	14y	F	3y	Chest tightness, shortness of breath, cough	XR: Unilateral infiltrate CT-T: Multiple, scattered granulomatous lesions	Abnormal	Non-caseating epithelioid granulomatous inflammation with giant cells	IFX	Rapid clinical and radiological complete resolution
Levenbrown [106]	2009	15y	F	Same time	Non-productive cough	XR: Bilateral patchy areas of opacification. CT-T: Bilateral peribronchovascular nodules and patchy infiltration	Abnormal	Non-caseating granulomatous inflammation with multinucleated giant cells, histiocytes, acute and chronic inflammation cell infiltrates. Organizing pneumonia.	PRD, MTX, MES	Mild improvement of PFT 4d after initiation of therapy. No comment on clinical improvement.
Mahgoub [107]	2007	5y	M	4y	Cough, shortness of breath	XR: Extensive airspace shadowing, ring shadows. CT-T: Interstitial lung disease	Not performed	Non-caseating epithelioid granulomatous inflammation, lymphocytic interstitial pneumonitis	1. Oral PRD 2. AZA	Only little overall improvement. Continuation of exercise-induced hypoxia. Lung transplantation was considered
Minic [108]	1998	15y	F	Same time	Exercise-induced dyspnea, cough	XR: Central density of the right middle lobe with a minor interlobular effusion	Normal	Subepithelial, non-caseating epithelioid granulomatous inflammation	Oral PRD, SUF	Resolution of symptoms after 2mo
Nelson [109]	2014	9y	F	2y	Cough, pleuritic chest pain, dyspnea on exertion	CT-T: Multiple subpleural lung nodules	Abnormal	Granulomatous inflammation, microabscesses with neutrophils and eosinophils	1. Oral PRD 2. IFX	Resolution of symptoms after 1mo. Relapse after discontinuation of PRD. Initiation of IFX led to complete resolution.
Pain-Prado [110]	2012	8y	F	5mo (b)	Tachypnea	XR: Interstitial infiltration CT-T: Multiple disseminated irregular stellar nodules	Normal	Not performed	No treatment	Complete resolution without treatment
Puntis [111]	1990	17y	M	2y	Unilateral pleuritic chest pain, productive cough, dyspnea	XR: Unilateral patchy consolidation, small pleural effusion	Not performed	Non-caseating epithelioid granulomatous inflammation	Isoniazid, Rifampicin	Complete resolution after 6w
Shah [112]	1976	13y	M	2y (b)	Cough	XR: Bilateral reticulonodular pattern	Not performed	Interstitial pneumonitis with multiple non-caseating granulomas	PRD, ferrous sulfate	Not mentioned
Silbermintz [113]	2006	13y	F	3y	Backpain, cough	XR: Bilateral infiltrates and densities CT-T: 2 rounded densities, multiple pleural and intraparenchymal pulmonary lesions	Abnormal	Non-caseating epithelioid granulomatous inflammation without interstitial fibrosis	IFX	Fast resolution of symptoms and improvement and XR
Taylor [18]	2020	10y	M	3y	None	CT-T: Bilateral pulmonary nodules.	Not performed	Non-necrotizing granulomatous bronchiolitis	Increased dose of ADA	Pulmonary nodules decreased
Taylor [18]	2020	17y	F	Same time	None	CT-T: Bilateral “tree-in-bud” opacities	Not performed	Necrotizing granulomatous inflammation and focal organizing pneumonia	Corticosteroids	Improvement on CT-scans
Taylor [18]	2020	14y	M	1mo	Non-productive cough	CT-T: Diffuse pulmonary micronodule	Not performed	Granulomatous inflammation	None	Symptom improvement shortly after starting maintenance therapy (IFX + MTX). Repeat CT-T 1y later with regression of pulmonary nodules
Taylor [18]	2020	14y	F	1y	Non-productive cough, pleuritic chest pain	XR: Right upper lobe nodule CT-T: Bilateral pulmonary nodules	Not performed	Chronic interstitial pneumonitis with patchy organizing pneumonia without granulomas	None	Symptom improvement. CT-T 8mo later with resolution of previous nodules, appearance of 2 new nodules
Vadlamudi [17]	2012	11y	F	Same time	Non-productive cough	XR: Unilateral consolidation and pleural effusion	Not performed	Not performed	PRD, IFX	Clinical and radiological improvement
Vadlamudi [17]	2012	17y	F	5y	Cough, shortness of breath	XR: Bilateral opacities CT-T: Bilateral cavitary lesions	Not performed	Not performed	IFX	Clinical and radiological improvement
Valletta [114]	2001	6y	F	5mo	Cough	XR: Unilateral parenchymal density	Normal	Thickening of basal membrane, angiectasis, active chronic inflammation	1. Antibiotics 2. PRD	Prompt resolution of symptoms with antibiotics, then after 2mo relapse with cessation of symptoms after short-term therapy with PRD

Abbreviations: δ, time between diagnosis of inflammatory bowel disease (IBD) and pulmonary manifestation (PM); y, years; mo, months; w, weeks; d, days; M, male; F, female; PFT, pulmonary function tests; (b), diagnosis of PM preceding IBD; XR, chest x-ray; CT-T, thoracic computed tomography; BSC, bronchoscopy; PRD, prednisolone; M, methylprednisolone; IFX, infliximab; ADA, adalimumab; AZA, azathioprine; MES, mesalamine; MTX, methotrexate; 6-MP, 6-mercaptopurine; SUF, sulfasalazine.

**Table 2 ijms-23-07287-t002:** Published cases of pulmonary manifestations in pediatric UC patients.

First Author	Publication Year	Age	Sex	δ	Pulmonary Symptoms	Radiographic Evaluation	PFT	Histopathology (Lung)	Therapy	Outcome
Basseri [115]	2010	17y	M	1y	Unilateral chest pain, shortness of breath	XR: Multiple bilateral patchy nodular opacities CT-T: Multiple bilateral nodules of varying sizes	Not performed	Multifocal acute alveolitis with multiple abscesses	PRD	Clinical and radiological improvement after 1mo
Carvalho [116]	2008	13y	F	1y	None	CT-T: 3 Nodules in lingula and left lower lobe	Abnormal	Marked alveolar inflammation, necrobiotic nodule	Intravenous steroids, parental nutrition, antibiotics, MES, 6-MP	Resolution of nodules in follow-up CT-T 12mo later
Gut [25]	2011	15y	F	1y	Productive cough, wheezing	XR: Mild bilateral hyperinflammation, bronchial thickening CT-T: Diffuse bronchiectasis of all lung fields	Abnormal	Not performed	Macrolides, high-dose steroids (budesonide), bronchodilators, physical therapy	Progressive improvement over 6mo
Krishnan [105]	2006	17y	M	4y	Chest pain	XR: Bilateral basal infiltrates and pleural effusion	Not performed	Bronchiolitis obliterans and organizing pneumonia	1. Oral PRD 2. IFX	Rapid clinical and radiological complete resolution
Mazer [117]	1993	13y	F	2y (b)	Non-productive cough, shortness of breath	XR: Bilateral diffuse interstitial infiltrate in a reticular pattern, peribronchial thickening, bronchiectasis, nodules	Abnormal	Necrotizing bronchiolitis and bronchiectasis with interstitial pneumonitis	1. Oral PRD 2. SUF	Improvement after initiation of SUF, relapse 8mo after discontinuation of PRD. Improvement after re-implementation of PRD
Russi [86]	2020	17y	F	6y	Chronic purulent cough	XR: Normal CT-T: “Tree-in-but” opacities, mild bronchiectasis, bronchial wall thickening	Abnormal	Not performed	MP, albuterol and inhalation with 3% hypertonic saline	Clinical improvement. Relapse 1y later with another therapy cycle and following improvement
Taylor [18]	2020	16y	M	1y	Pleuritic chest pain	XR: Bilateral pulmonary opacities CT-T: Multiple pulmonary nodules	Not performed	1. Biopsy: Necrotic neutrophilic nodules without granulomas	1. PRD 2. IFX	Initial resolution of symptoms. Recurrence 1y later during IBD flare, with dramatic improvement after initial of IFX
Taylor [18]	2020	13y	F	1y	Chronic productive cough, exertional dyspnea	XR: Multiple pulmonary nodules CT-T: Multiple pulmonary nodules, organizing pneumonia	Abnormal	Cryptogenic organizing pneumonia without granulomas	Fluticasone and albuterol inhalation	CT-T 2mo later with improvement of nodules and no evidence of pneumonia. Improvement of cough, but continuous dyspnea on exertion
Teague [118]	1985	12y	M	5y	Exertional dyspnea	XR: Extensive, bilateral nodular interstitial pattern	Abnormal	Desquamative interstitial pneumonitis	1. Oral PRD 2. AZA	No sufficient improvement with finally fatal respiratory failure 2y after diagnosis

Abbreviations: δ, time between diagnosis of inflammatory bowel disease (IBD) and pulmonary manifestation (PM); y, years; mo, months; M, male; F, female; PFT, pulmonary function tests; (b), diagnosis of PM preceding IBD; XR, chest x-ray; CT-T, thoracic computed tomography; PRD, prednisolone; MP, methylprednisolone; IFX, infliximab; MES, mesalamine; 6-MP, 6-mercaptopurine.

## Data Availability

Not applicable.

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
