# Peer review of "Pathophysiological Concepts and Management of Pulmonary Manifestation of Pediatric Inflammatory Bowel Disease"

_ijms, 2022, doi:10.3390/ijms23137287_

Round 1
Reviewer 1 Report
This review gives a detailed summary of the pathophysiology concepts of PM and the important clinical issues in pediatric IBD. I read this review with great interest because it provide guidance for readers to understand the immunological/molecular mechanism of PM in IBD and its clinical management, which will be benefit for the prevention and mitigation of progressive lung damage in adult IBD. Here are my comments regarding this manuscript:
1. Fecal microbiota transplantation (FMT) is an effective treatment for IBD. FMT-related lung injury has not been reported at present. The important role of gut microbiota in the “gut-lung axis” is being explored, and FMT shows the potential benefits for lung infection, asthma and others. Therefore, FMT should be considered for the treatment of IBD patients with medication-induced lung injury.
2. It is inappropriate to list mesalamine [MES] and sulfasalazine [SUF] as immunosuppressives.
3. Many grammatical problems and spelling errors have been found and the author should revise them carefully. Some errors are listed below:
3.1. “UC” not “CU”
3.2. Nc margination (line 105)?
3.3. “homeostasis” not “hemostasis” (line 131)
3.4. “vicious” not “vitious” (line 180)
3.5. “and” not “unt” (line 459)
3.6. The temporal contradiction exists in multiple sentences, such as line 161-163 (“were detected” should be deleted), line 188-189, line 211 and so on.
3.7. Line 216: “was” should be deleted.
3.8. Line 126: “and” should be deleted.
3.9 Line 107-109: “not larger in diameter post-capillary venules” ??
……
Author Response
Reviewer #1: “This review gives a detailed summary of the pathophysiology concepts of PM and the important clinical issues in pediatric IBD. I read this review with great interest because it provide guidance for readers to understand the immunological/molecular mechanism of PM in IBD and its clinical management, which will be benefit for the prevention and mitigation of progressive lung damage in adult IBD.”
Author’s Response: We thank reviewer #1 for the positive feedback.
Reviewer #1: “Fecal microbiota transplantation (FMT) is an effective treatment for IBD. FMT-related lung injury has not been reported at present. The important role of gut microbiota in the “gut-lung axis” is being explored, and FMT shows the potential benefits for lung infection, asthma and others. Therefore, FMT should be considered for the treatment of IBD patients with medication-induced lung injury.”
Author’s Response: We thank reviewer #1 very much for this important aspect regarding new therapy options for IBD patients. Fecal microbiota transplantation (FMT) has indeed shown some promising effect on induction of remission in UC patients. We found one FMT-related lung injury (pneumonia), which was reported during a randomized controlled trial by Costello et al. (J Crohn´s Colitis 2017;11:S23; doi:10.1093/ECCO-JCC/JJX002.035). We added this aspect to the chapter “Medication-associated lung injury”.
Reviewer #1: “It is inappropriate to list mesalamine [MES] and sulfasalazine [SUF] as immunosuppressives.”
Author’s Response: We appreciate this helpful comment by reviewer #1 regarding the wrong categorization of mesalamine and sulfasalazine as immunosuppressives. We corrected this sentence.
Reviewer #1: “Many grammatical problems and spelling errors have been found and the author should revise them carefully. Some errors are listed below: “UC” not “CU”; Nc margination (line 105)?; “homeostasis” not “hemostasis” (line 131); “vicious” not “vitious” (line 180); “and” not “unt” (line 459); the temporal contradiction exists in multiple sentences, such as line 161-163 (“were detected” should be deleted), line 188-189, line 211 and so on; line 216: “was” should be deleted; line 126: “and” should be deleted; line 107-109: “not larger in diameter post-capillary venules” ?? ……”
Author’s Response: We want to thank you very much for the careful reading of our manuscript and that you found the mentioned grammatical and spelling errors. We overlooked our manuscript again and corrected all of them.

Reviewer 2 Report
This review article summarizes current knowledge about PM in a rare condition in pediatric IBD patients, focusing on pathophysiological and clinical aspects. Through pathophysiological concepts, medical imaging tests, and treatment research etc.in clinical illustrated the potential mechanism of PM in IBD children’s patients. At the same time the author proposed a harsh opinion that is “Regarding the question, if PFT are a good tool for identifying PM in children, current literature in contradictory.”. This question deserves more thought by staff engaged in related research or can provide ideas for further in-depth research on PM IN IBD PATIENTS in the future.
The whole text is clearly thought out, the collected materials are sufficiently reasonable, the analysis is reasonable and well-founded, and can finally put forward reasonable clinical recommendations for the treatment of drugs.
Recommend, published!
Author Response
Reviewer #2: “This review article summarizes current knowledge about PM in a rare condition in pediatric IBD patients, focusing on pathophysiological and clinical aspects. Through pathophysiological concepts, medical imaging tests, and treatment research etc. in clinical illustrated the potential mechanism of PM in IBD children’s patients.”
Author’s Response: We thank reviewer #2 for the positive feedback.
Reviewer #2: “At the same time the author proposed a harsh opinion that is “Regarding the question, if PFT are a good tool for identifying PM in children, current literature in contradictory.”. This question deserves more thought by staff engaged in related research or can provide ideas for further in-depth research on PM IN IBD PATIENTS in the future.”
Author’s Response: We thank reviewer #2 very much for this helpful comment on our statement regarding the use of PFT in the conclusion chapter. We modified this comment, having now a more positive meaning to it. Furthermore we added the results of a study of Furlano et al. (Respiration 2015;90:279–286), which showed some significant alterations in PFT in pediatric IBD patients.
Reviewer #2: “The whole text is clearly thought out, the collected materials are sufficiently reasonable, the analysis is reasonable and well-founded, and can finally put forward reasonable clinical recommendations for the treatment of drugs.”
Author’s Response: We thank reviewer #2 for the assessment.
Reviewer #2: “Recommend, published!”
Author’s Response: Thank you.
